

# The pitfalls of short-range endemism: high vulnerability to ecological and landscape traps

Leanda D. Mason[1], Philip W. Bateman[1] and Grant W. Wardell-Johnson[2]

[1] School of Molecular and Life Sciences, Curtin University, Bentley Campus, Perth, WA, Australia
[2] ARC Centre for Mine Site Restoration, School of Molecular and Life Sciences, Curtin University, Bentley Campus, Perth, WA, Australia

## ABSTRACT

Ecological traps attract biota to low-quality habitats. Landscape traps are zones caught in a vortex of spiralling degradation. Here, we demonstrate how short-range endemic (SRE) traits may make such taxa vulnerable to ecological and landscape traps. Three SRE species of mygalomorph spider were used in this study: *Idiommata blackwalli*, *Idiosoma sigillatum* and an undescribed *Aganippe* sp. Mygalomorphs can be long-lived (>43 years) and select sites for permanent burrows in their early dispersal phase. Spiderlings from two species, *I. blackwalli* ($n = 20$) and *Aganippe* sp. ($n = 50$), demonstrated choice for microhabitats under experimental conditions, that correspond to where adults typically occur *in situ*. An invasive veldt grass microhabitat was selected almost exclusively by spiderlings of *I. sigillatum*. At present, habitat dominated by veldt grass in Perth, Western Australia, has lower prey diversity and abundance than undisturbed habitats and therefore may act as an ecological trap for this species. Furthermore, as a homogenising force, veldt grass can spread to form a landscape trap in naturally heterogeneous ecosystems. Selection of specialised microhabitats of SREs may explain high extinction rates in old, stable landscapes undergoing (human-induced) rapid change.

# INTRODUCTION

Habitat-selection theory suggests behavioural and life history decisions prompted by environmental cues act as indicators of habitats that maximise fitness (*Rosenzweig, 1981*). Misguided habitat choice may be based on environmental cues associated with high-quality habitats but that are no longer reliable in disturbed or changing environments. Choices based on such cues lead to organisms entering 'ecological traps' (*Delibes, Gaona & Ferreras, 2001*; *Dwernychuk & Boag, 1972*). Ecological traps pose a serious conservation concern for a wide range of taxa (*Battin, 2004*; *Hale & Swearer, 2016*; *Hale, Treml & Swearer, 2015*). A review of ecological traps shows that historically, most studies have focused on highly mobile bird species and their choice of nesting sites but more recently there has been attention to other, commercially relevant, taxa (*Robertson, Rehage & Sih, 2013*). Here we argue that ecological traps are

Corresponding author
Leanda D. Mason,
leanda.mason@curtin.edu.au

particularly threatening for short-range endemic (SRE) taxa. Persistence of these taxa is further compromised in disturbed landscapes for two main reasons (1) their specialist co-evolutionary requirements, and (2) limitations in their capacity to relocate following disturbance.

Highly disturbed landscapes form 'landscape traps', compromising ecological function and compounding the influence of ecological traps for many taxa (*Lindenmayer et al., 2011*). For example, logging and altered fire regimes can interact in ways that cause the irreversible modification of ecological function. Landscape traps may also be associated with negative feedback loops between fire and weed invasion (*Fisher et al., 2009a*). Thus, fire provides opportunity for weed invasion, which then provides greater fuel loads, increasing the capacity for more intense fires.

Regions with high proportions of endemic taxa represent a unique and ongoing challenge in conservation biology worldwide (*Wardell-Johnson & Horwitz, 1996*). Landscape age may explain high endemism and biodiversity in some sites (*Mucina & Wardell-Johnson, 2011*) with complex microhabitats and niche partitioning having established and evolved over time in stable conditions (*Rosenzweig, 1981*). Old, stable landscapes may therefore be more susceptible to landscape traps due to increased numbers of associations—providing a higher chance of disturbances that lead to negative feedback loops (*Lindenmayer et al., 2011*). Historical divergence in old, stable landscapes may be primarily attributed to two major factors (1) heterogeneously distributed habitat and narrow, subdued geographical barriers (*Wardell-Johnson & Horwitz, 1996*), and (2) taxa with limited mobility and dispersal capabilities (*Rix et al., 2014*).

A taxon with a distribution of less than 10,000 km$^2$, low reproductive output and limited mobility and dispersal fulfills the criteria for being a SRE taxon (*Harvey, 2002*). SREs are of high conservation concern as a restricted distribution contributes to rarity and increased chance of extinction (*Harvey, 2002*; *Rabinowitz, 1981*). Vulnerability is exacerbated when combined with a limited capacity to adapt or escape human-induced rapid environmental change (HIREC) (*Sih, Ferrari & Harris, 2011*). SREs tend to be specialists, and representation is proportionally higher in invertebrates than vertebrates (*Harvey, 2002*). Old, stable landscapes are likely to have the highest numbers and proportions of SREs globally (*Rix et al., 2014*).

The ancient and highly heterogeneous landscape of south-western Australia (SWA) hosts many SRE species (*Rix et al., 2016*). By their nature, SREs are typically rare and cryptic, making it difficult to gather essential information for conservation decisions. Poor dispersal capabilities suggest that SREs are able to survive in small patches of high-quality remnant vegetation (*Mason, Wardell-Johnson & Main, 2016*). Mygalomorph spiders are excellent candidate taxa as bioindicators for SRE taxa. They are sedentary, meaning individuals can be monitored long-term. Choice of microhabitat by a dispersing spiderling is a vital first step to successful investment in a life-long burrow. Burrows are investments in terms of effort and time to establish. It is rare for mygalomorphs to move as adults, as such moves come at high energy cost and leave them vulnerable to predation and desiccation (*Mason et al., 2013*).
As mygalomorph matriarchs can live up to 43 years (*Mason, Wardell-Johnson & Main, 2016*), appropriate burrow-site and habitat choice by spiderlings is vital.

We aimed to determine whether mygalomorph spiderlings of three species display choice of microhabitat conditions that reflects their realised niche as adults. Realised niche was recorded as adult burrow microhabitats that were regularly recorded during an extensive survey. Burrow microhabitats of adults of each species were used to inform the microhabitat habitat preference choices provided to all spiderling species. *Mason, Wardell-Johnson & Main (2016)* suggested that invasive veldt grass (*Erharta* spp.) may be detrimental to local mygalomorph species, either through preventing establishment of burrows, or through microhabitat or prey availability variables. Veldt grass was therefore made available as one of four ground cover choices for mygalomorph spiderlings in a controlled field trial. Microhabitat variables of relative humidity (RH), ambient temperature (Ta) and prey species assemblages were also measured to explain spiderling choice.

## METHODS

### Study site

This study was conducted in Perth, a highly urbanised centre in the Mediterranean-climate zone (mean annual rainfall 740 mm) of SWA. This highly biodiverse region has lost almost 90% of the original habitat due to extensive deforestation and on-going disturbance, qualifying it as a global biodiversity hotspot (*Myers et al., 2000*; *Wardell-Johnson et al., 2016*).

### Study species

We collected an individual *Idiommata blackwalli* (Barychelidae—brush-footed trapdoor spiders, body length: 32 mm) with a brood size of 42 and individuals of two genera of Idiopodae (armoured trapdoor spiders); an *Aganippe* sp. (body length: 27 mm) and an *Idiosoma sigillatum* (body length: 36 mm) which had broods of 65 and 78, respectively. The idiopid spiderlings were approximately 3 mm in length and those of *I. blackwalli* 7 mm long.

Despite large superficial differences between *Aganippe* and *Idiosoma*, genetic techniques have recently found these genera to be closely related. The extreme sclerotisation of the abdomens of *I. sigillatum* likely relates to behaviour where they block their burrow against intruders. Burrow morphology supports this behaviour by having a diagonostic 'atrium' where the spider can manoeuvre near their lid burrow during foraging, but tightly constricts down the shaft. Foraging range is thought to be enhanced for *I. sigillatum* due to their burrows having twig-line attachments; this enables them to detect prey species along the twigs via vibrations. Both *Aganippe* and *Idiosoma* are highly defensive of their burrows, retreating to the bottom and rearing when disturbed. Conversely, the *I. blackwalli* collected was quite docile by comparison and easily observable in her shallow (15 cm deep) bulb-shaped burrow. Photos of each spider species and their typical burrow entrance has been provided for comparison (Fig. 1).

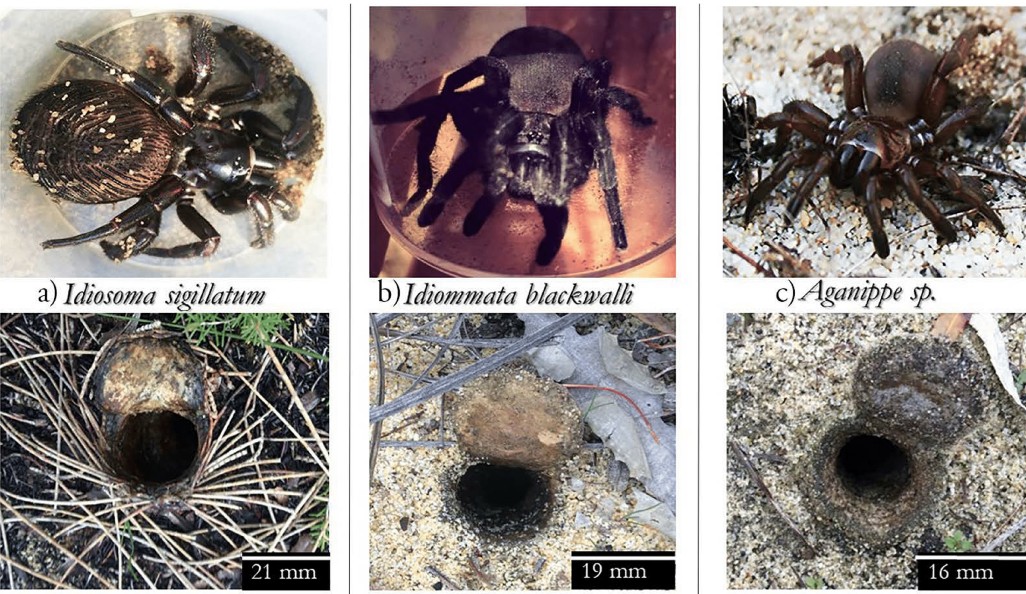

**Figure 1 Subject spider species and their burrows.** Three species of mygalomorph spider used in the study and their corresponding burrows; (A) *Idiosoma sigillatum*, (B) *Idiomatta blackwalli* and (C) *Aganippe sp*. Scale bars in the bottom right corner of burrow photos refers to the widest point across the burrow entrance. All three species occur within patches of urban bushland within Perth, Western Australia. Photo credits: Bill Bateman for *Aganippe sp*. spider, Jessica Moran for *Idiommata blackwalli* spider and Leanda Denise Mason for all others.

## Microhabitat preference

Adult spider burrows were located during an extensive survey of Perth urban remnant vegetation that included 135 quadrats (100 m × 100 m) in 41 sites (see *Mason, Wardell-Johnson & Main, 2016* for further details). Microhabitat variables to included leaf litter cover (%) and leaf litter type within a one metre square quadrat centred on burrows.

To reflect the natural dispersal time of mygalomorph species a habitat preference experiment was conducted, using the collected spiderlings, from 24 April to 24 May in the field trials area at Curtin University (Bentley campus). A total of 50 circular enclosures 10 cm high and 30 cm in diameter were constructed from clear plastic, sunk 3 cm into the sand and sealed with cling-wrap to deter spiderling escape. Four microhabitat treatments were provided within these enclosures; bare soil, veldt grass tufts, *Allocasuarina fraseriana* (Casuarinaceae) and *Banksia attenuata* (Proteaceae) litter. Within each enclosure, configuration of microhabitats was divided into eighths with the two replicate treatments positioned directly opposite one another. An individual spiderling was released in the centre of each enclosure and the location of their burrow recorded on the following day. Chi-square tests were performed for each species using StatistXL v.1.9 to identify microhabitat preference, both in adults and dispersing spiderlings.

## Microhabitat characteristics

Ambient temperature and relative humidity was recorded hourly using iButtons in each of the four microhabitat types in the field: bare ground, veldt grass tufts, leaf litter of
**Table 1** Microhabitat preferences for three mygalomorph spider species (*Aganippe* sp., *Idiommata blackwalli* and *Idiosoma sigillatum*).

| | Species | Allocasuarina | Banksia | Bare ground | Veldt grass | n | d.f. | p-Value |
|---|---|---|---|---|---|---|---|---|
| *Spiderlings* | *Aganippe* sp. | 8 | 12 | 21 | 9 | 50 | 3 | 0.04 |
| | *Idiommata blackwalli* | 3 | 11 | 3 | 3 | 20 | 3 | 0.01 |
| | *Idiosoma sigillatum* | 2 | 0 | 0 | 38 | 40 | 3 | <0.001 |
| *Adults* | *Aganippe* sp. | 1 | 2 | 10 | 0 | 13 | 3 | <0.001 |
| | *Idiommata blackwalli* | 3 | 15 | 0 | 0 | 18 | 3 | <0.001 |
| | *Idiosoma sigillatum* | 17 | 4 | 4 | 0 | 25 | 3 | <0.001 |

**Note:**
Spiderling preference was recorded under controlled experimental conditions in Perth, south-western Australia. The number of adult burrows found in situ ground cover microhabitats (*Allocasuarina, Banksia*, bare ground, and veldt grass) is shown.

*A. fraseriana* and leaf litter of *B. attenuata*. A total of 10 iButtons were placed in each microhabitat type at two remnant bushland study sites (Kings Park: 31°57′46.0″S 115°49′52.8″E and Bold Park: 31°56′53.1″S 115°46′13.6″E) and at the spiderling experiment field trial area (32°00′43.1″S 115°53′23.1″E). All iButtons were placed where leaf litter and top layer of soil interacted, except for bare soil microhabitats where iButtons were placed on top of the soil. Records of Ta and RH during 24 May to 6 June 2016 were used for analysis.

To assess abundance and diversity of potential prey species, we collected 20 *in situ* ground-cover samples for each of the four microhabitats; 10 from Bold Park and ten from Kings Park. Samples were collected within a 20 cm × 20 cm quadrat and were pooled from the same microhabitat and site. Samples were placed in a light-funnel for a week and leaf-litter invertebrates were collected in a jar filled with 100 mL of ethylene glycol. These invertebrates were later categorised by size and recognisable taxonomic units (RTUs) (*Oliver & Beattie, 1993*).

# RESULTS

## Microhabitat preference

Significant differences in microhabitat preference were found for all three species. Adult microhabitat corresponded with spiderling microhabitat preferences for both *Aganippe* sp. and *I. blackwalli*; spiderlings of these two species preferred open ground cover and *B. attenuata* leaf litter respectively (Table 1).

*Idiosoma sigillatum*, a species that typically attaches twig-lines to the rim of their burrow, were predominately found in *A. fraseriana* leaf litter as adults, but selected veldt grass almost exclusively as spiderlings (Table 1). In these cases, no twig-lining was apparent, with one exception where *Allocasuarina* litter from an adjacent treatment was used.

## Microhabitat characteristics

A comparison between microhabitat Ta revealed little difference, with all averaging between 14 and 15 °C. Variability in Ta remained most stable in *B. attenuata* leaf litter (range: 23.0, SD: 3.9), followed by veldt grass (range: 23.2, SD: 4.2), *A. fraseriana* leaf litter (range: 23.3, SD: 4.5) and bare ground (range: 26.0, SD: 5.3) (Fig. 2).

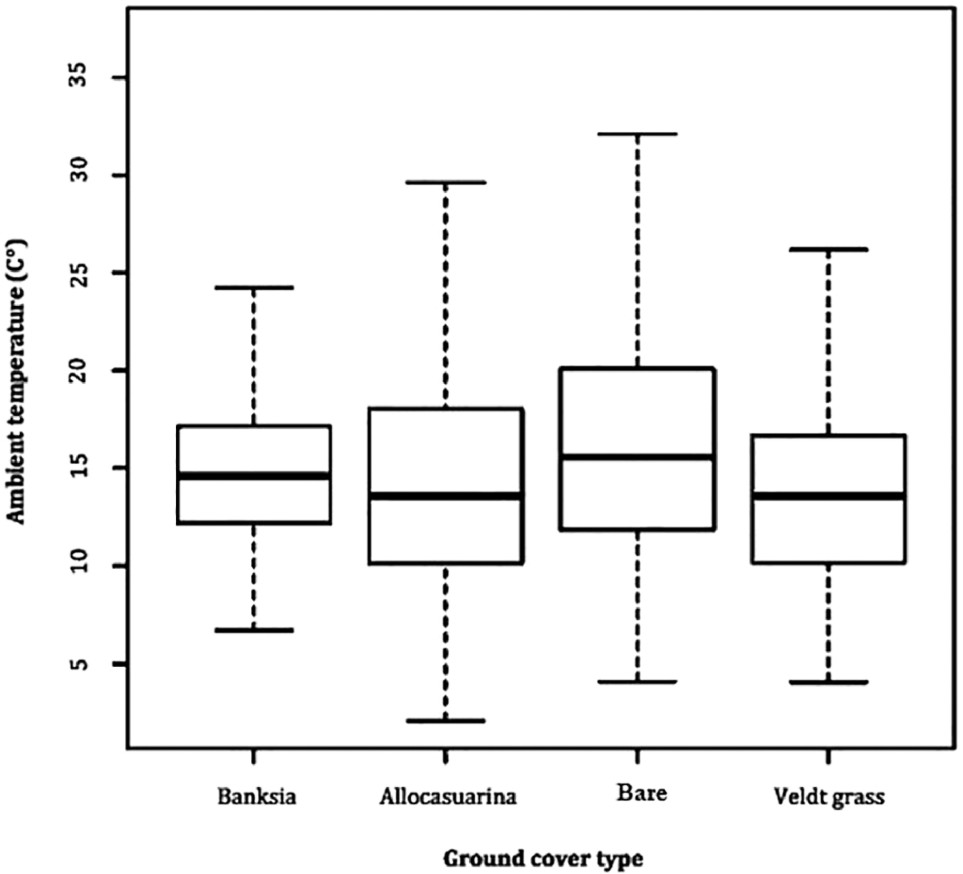

**Figure 2 Ambient temperature fluctuation in microhabitats.** A box-and-whisker plot comparing the differences in ambient temperature (°C) fluctuation during 24 May to 6 June, 2016 in different ground cover types (*Banksia* leaf and *Allocasuarina* leaf litter, bare ground and veldt grass) found in urban bushland of Perth, Western Australia.

Relative humidity exposed far greater difference between microhabitats. Leaf litter of *B. attenuata*, leaf litter of *A. fraseriana* and open ground cover all averaged 87.2–87.4%. However, a higher RH (93%) was found in veldt grass. Veldt grass retained the most moisture (range: 51.7, SD: 10.5), followed by leaf litter of *B. attenuata* (range: 62.9, SD: 12.6), leaf litter of *A. fraseriana* (range: 43.8, SD: 10.6) and open ground (range: 72.2, SD: 15.5) (Fig. 3).

The size of potential prey species varied greatly between microhabitat types. Size classes of 1 and 10 mm yielded high numbers (i.e. <100 individuals) across all microhabitats and can be attributed primarily to two RTUs being Collembola (springtails). Veldt grass yielded the least potential prey in both abundance and species richness (n: 213, S: 10). By comparison, open ground had greater numbers (n: 483) but similar species richness (S: 11) to veldt grass, while *Banksia* leaf litter provided more species (S: 21), but lower abundance (n: 393) than open ground. The most productive ground cover was *A. fraseriana* with both highest abundance (n: 6591) and richness (S: 23) (see Supplementary Materials).

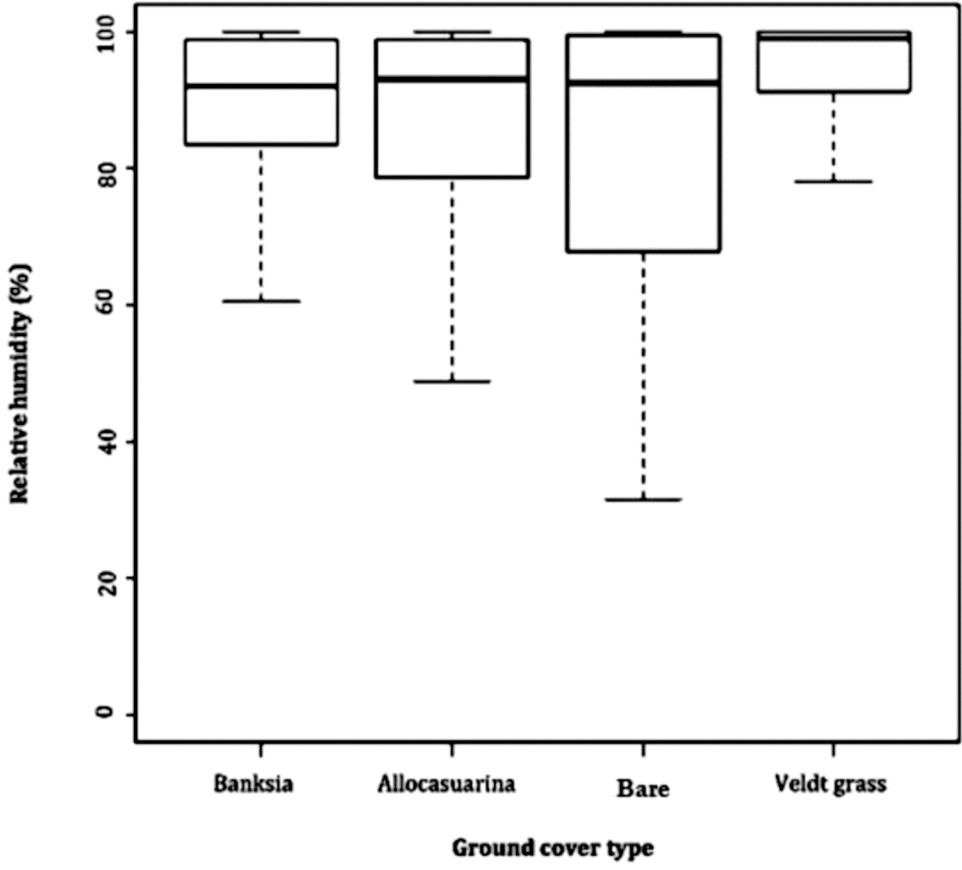

**Figure 3 Relative humidity fluctuations in different microhabitats.** A box-and-whisker plot comparing the differences in relative humidity (%) fluctuation during 24 May to 6 June, 2016 in different ground cover types (*Banksia* leaf and *Allocasuarina* leaf litter, bare ground and veldt grass) found in urban bushland of Perth, Western Australia.

## DISCUSSION

Habitat choices consistent with an ecological trap were revealed by *I. sigillatum* spiderlings, as they showed a high preference for invasive veldt grass over native *A. fraseriana* leaf litter (where adults were typically found). *Idiosoma sigillatum* spiderlings appeared to have fallen into an ecological trap, as indicated by five factors. Firstly, a high RH in veldt grass may have enticed the spiderlings into this microhabitat. High RH may be an evolutionary trap for this species, as high RH is a physiological requirement for mygalomorphs (*Mason et al., 2013*). Secondly, *I. sigillatum* is one of a few mygalomorph species in SWA that have a 'moustache' twig-line arrangement attached to their burrow rim. This is apparent even as spiderlings when establishing in *A. fraseriana* leaf litter. Foraging capabilities may therefore be further compromised when establishing in veldt grass. Thirdly, previous studies have indicated that veldt grass compromises ecosystem functioning (*Fisher et al., 2006*) and this appears to be supported by our data on low prey species abundance and diversity in veldt grass in comparison to *A. fraseriana*. Fourthly, if, in the wild, *I. sigillatum* spiderlings disperse into veldt grass patches, then there is no evidence they have persisted to adults.
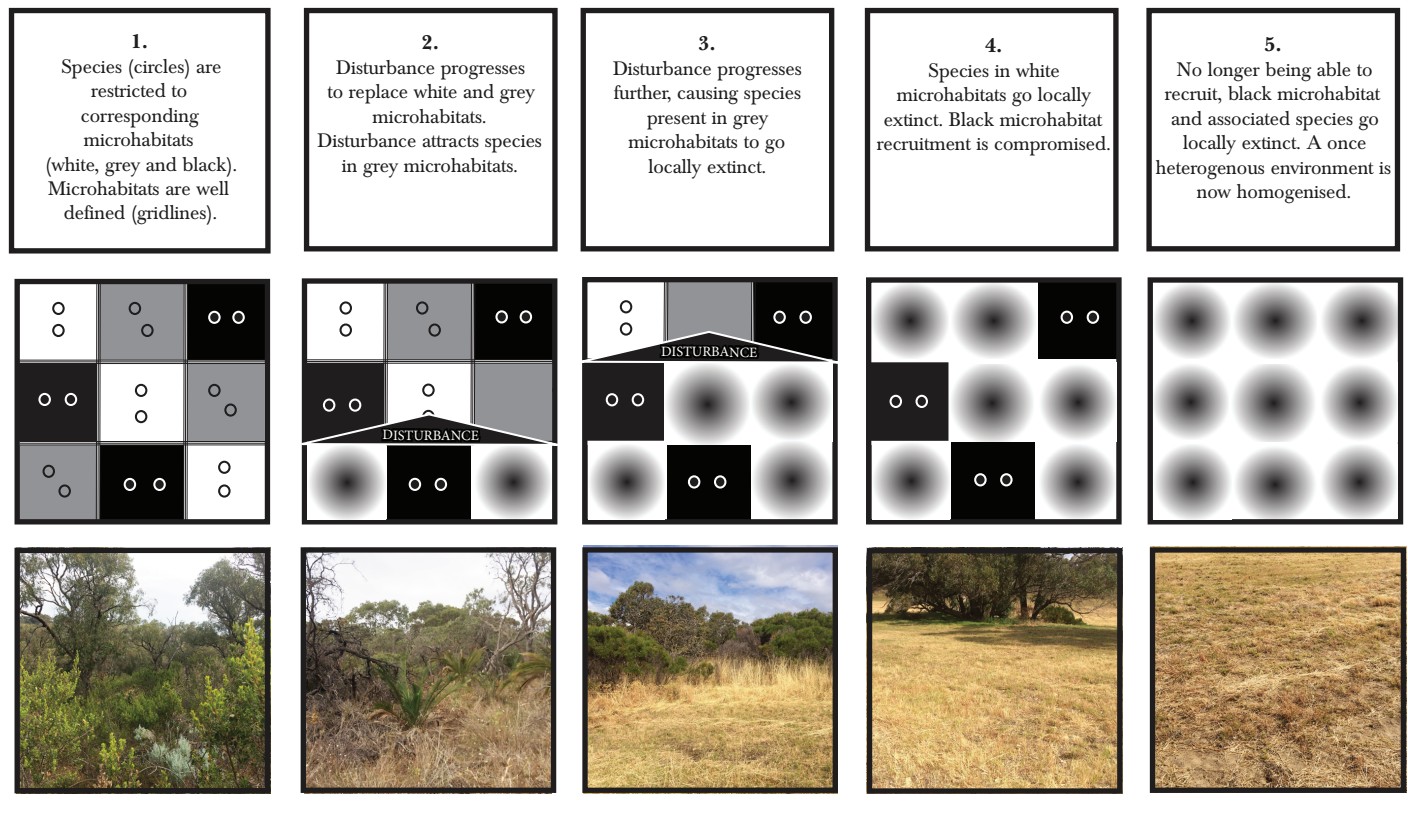

**Figure 4 Short-range endemics get caught in ecological and landscape traps.** Hypothetical representation of short-range endemic species in response to ecological traps and landscape traps over time. The example, including all photographs, is from Bold Park in Perth metropolitan area of Western Australia. Photo credit: Leanda Denise Mason.

Lastly, should spiderlings initially establish in veldt grass and then move, this would result in high energy costs, and leave them vulnerable to predation and desiccation.

Mygalomorph spiders have exceptionally low fecundity, rendering the probability of locating a brooding mygalomorph spider mother during single surveys very low. Therefore, we were exceptionally fortunate to conduct this study using three individual broods from three species. While there is potential for maternal effects on habitat preference, life history and realised niches of mygalomorph adults suggest a low likelihood of a change in outcome. Our confirmation of mygalomorph spiderling choice in microhabitats is novel. We therefore suggest investigation in other mygalomorph spiders with ready access to brood, such as tarantulas that are routinely used in the pet trade.

The impact of invasive grasses effectively transforms and homogenises the ecosystem. By facilitating and interacting with other transforming processes (e.g. nutrient status and fire regime) these grasses form a landscape trap (*Fisher et al., 2009b*). Reliance on microhabitats, as demonstrated by *Aganippe* sp. and *I. blackwalli*, and other SRE species (*Harvey et al., 2011*), means that the spread of veldt grass to form a landscape trap is likely to be a threatening process (Fig. 4).

As an invasive weed, veldt grass tends to homogenise environments by out-competing native species (*Fisher et al., 2009b*) and replacing microhabitats. In highly heterogeneous environments that harbour a high diversity of specialist, endemic and co-evolved taxa, homogenisation poses a significant threat to biodiversity (*Wardell-Johnson & Horwitz, 1996*). Homogenisation of previously heterogeneous habitats is the underlying cause for biodiversity loss worldwide. This has particular significance for the transformation of native habitats through human land-use practices (*Sih, Ferrari & Harris, 2011*). In terms of effective management, re-introduction of native species into habitats after a homogenising disturbance would be a poor use of resources (*Fisher et al., 2009b*). Preventative rather than restorative measures are required (*Wardell-Johnson et al., 2016*). Restoration is impracticable for the level and time required to rebuild complexity and associated microhabitats (*Rebelo et al., 2011*). The prevention of likely ecological traps and determination of the circumstances that transform them into landscape traps is imperative for conservation decision-making.

Sedentary taxa enable reliable monitoring of individuals and populations, generating invaluable data for conservation decision-making (*Main, 1978*). The majority of mygalomorph species exhibit the SRE traits of low mobility, low dispersal, low fecundity, slow growth, restriction to particular habitats, and entire distribution being less than 10,000 km$^2$ (*Harvey, 2002*). Some mygalomorph species have capacity to balloon (*Pétillon et al., 2012*), but this may have evolved to mitigate territorial behaviour (L. D. Mason, 2017, unpublished data) rather than to increase dispersal capability. As such, the biology of particular SRE groups will still need to be considered in management for conservation. Mygalomorphs provide an excellent exemplar of how other SRE may react to novel threatening processes. Unfortunately, exceptionally long lifespan and low reproductive output mean that reduction in fitness, through habitat clearance, disturbance and degradation e.g. through weed invasions, may already be causing localised extinctions in mygalomorphs and other SREs (*Rix et al., 2016*).

We found invasive veldt grass to be a poor-quality microhabitat compared to native microhabitats. Veldt grass provides an example of ecological and landscape traps in which SRE taxa, such as mygalomorphs, may be caught. In a global context, all specialist species that cannot readily re-locate or adapt are particularly vulnerable to the formation of landscape traps through homogenisation of native microhabitats. While broad-scale management would be ineffective in heterogeneous environments (*Wardell-Johnson & Horwitz, 1996*), targeting specific, widespread traps, such as weed invasion, may control homogenisation and consequent extinctions.

## CONCLUSION

Informed justification allows cost-effective investment in conservation decision-making. HIREC is a significant threat to biodiversity in old, stable landscapes, which are prevalent globally (*Mucina & Wardell-Johnson, 2011*). Ecosystem functioning will inevitably be compromised and SRE species will go extinct without appropriate conservation management. Identifying potential ecological and landscape traps early will allow us to better manage SREs and other specialist taxa prevalent in heterogeneous

habitats. Further studies into the interactions between taxa traits and disturbance will allow more effective management, potentially curtailing impacts of current threatening processes.

### Funding

Funding was supplied through the Australian Postgraduate Award at Curtin University, Bentley campus. There was no additional external funding received for this study. The funders had no role in study design, data collection and analysis, decision to publish, or preparation of the manuscript.

### Grant Disclosures

The following grant information was disclosed by the authors:
Australian Postgraduate Award at Curtin University, Bentley campus.

### Competing Interests

The authors declare that they have no competing interests.

### Author Contributions

- Leanda D. Mason conceived and designed the experiments, performed the experiments, analyzed the data, contributed reagents/materials/analysis tools, prepared figures and/or tables, authored or reviewed drafts of the paper, approved the final draft.
- Philip W. Bateman conceived and designed the experiments, authored or reviewed drafts of the paper, approved the final draft, contributed ideas on 'ecological traps'.
- Grant W. Wardell-Johnson conceived and designed the experiments, authored or reviewed drafts of the paper, approved the final draft, contributed ideas on 'landscape traps'.

### Data Availability

The raw data are provided in the Supplemental File.

### Supplemental Information

Supplemental information for this article can be found online at http://dx.doi.org/10.7717/peerj.4715#supplemental-information.

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
