# Peer review of "The pitfalls of short-range endemism: high vulnerability to ecological and landscape traps"

_PeerJ, doi:10.7717/peerj.4715_

## Round 0.1 · original submission · Major Revisions

Please read through the comments and address each suggested change either in the manuscript or explain why it was not incorporated.

Reviewer 1 ·

Basic reporting

The article was clear and well-written. I found no grammatical errors.

Mason et al provide sufficient background information for their study in their introduction, although they may consider putting their study in a wider context given that there are many arachnid taxa around the world that are short range endemics.

The authors should consider providing more information about the individual species in their study. Is there available life history information in the species descriptions that are relevant for comparing and contrasting the results shown here?

The Figure and Table were informative and clear. The authors provide summary results for their measurements of temperature and humidity, as wells as prey abundance and diversity, but they should provide additional tables with more information for these. I found the information regarding the prey taxa and size particularly lacking. The authors should provide the raw data for their collection and categorization of prey items in a supplementary table.

Experimental design

Mason et al present novel findings in their study that will be of value to researchers who study short range endemic taxa. Mygalomorphs are logistically a very difficult group to work with for behavioral studies due to their cryptic nature and longevity. General observations of microhabitat specificity have been made for mygalomorphs, but Mason et al present the first statistical tests of spiderling habitat choice for burrow construction. As the authors state, this has important conservation implications given that mygalomorphs are low-vagility taxa, as mygalomorph spiders would have to expend considerable energy to seek a new place to generate a burrow.

Mason et al have conducted a thorough study with only a few exceptions. The sample size Mason et al use is small given that only three species and only one brood from each of these species is studied. However, given the difficulty of collecting mygalomorphs in the wild, especially mygalomorph spiderlings, this is understandable. The statistical analyses for spiderling choice are robust and convincing. However, only a few descriptive statistics are offered for the temperature and humidity tests, and no tests are offered to show if there is a significant difference between microhabitat types. The same is true for their tests of abundance and diversity of prey types in different microhabitats. It is also not clear what categories Mason et al used for size and what exactly they mean by recognizable taxonomic unit, thus making this aspect of their study difficult to replicate.

Validity of the findings

As stated above, the conclusions drawn from the test of spiderling choice are robust and convincing. However, the lack of tests of significant differences for environmental and prey item measurements limits the authors’ ability to draw conclusions about the role of these variables in microhabitat choice. That said, the discussion is well-written, convincing, and addresses the original research question. I found limited, if any, speculation.

Additional comments

The authors should consider providing a picture of each species, if available.

Can the authors comment on whether the same enclosure was used for multiple spiderlings in their tests of microhabitat choice? If this is the case, did the authors attempt to prevent any influencing factors among spiderlings? For example, were observed silk threads removed that could potentially influence (in a positive or negative orientation) where later spiderlings might place their burrow?

The authors should consider commenting on aspect as it relates to veldt grass coverage. Mygalomorph spiders typically prefer shaded slopes. Did the authors observe veldt grass more commonly on south-facing, shaded slopes, which might be particularly detrimental for mygalomorphs?

Reviewer 2 ·

Basic reporting

English is good, writing is clear. Bibliography and structure of the ms are both correct.

To me the issue comes from the gap between the objective of the study, the hypotheses and the results.

This study basically compares the (micro-)habitat preference of adults vs. juveniles of three mygalomorph species. It consists, at best (because methods are differ between stages, making results hardly comparable), in comparing habitat selection among species with different life history. The assessment of the impact of an invasive plant of the mygalomorphs, which could be an alternative story, is even not possible given the data available (see detailed comments below). To sum up, the study cannot be published as such, or at least not regarding the objectives and the logics it has. I’m sorry to be so tough, but it is an extremely descriptive work, poorly replicated, with very few experimental data and lacking many details. I hope the comments can still be helpful for a further submission.

Experimental design

Many details are missing, rendering the study impossible to replicate.

Abstract: Sampling methods, experimental and field works, names of the species / families, number of samples / individuals, location and date of the study need to be added.
L74-77: Did you use field or lab’ work? Did you conduct surveys or experiments?
L105-107: how did you standardize the litter? By volume, weight? Did you dry it before?
L109-110: Where and when did you do the lab’ work?
L129: How did you define the RTUs? Which books / keys did you use?

Validity of the findings

I’m afraid the authors the proper design regarding what you want to assess (e.g. would larger-range species react differently?). The study should be re-written with another objective, i.e. comparing the local distribution of different spider species.

L3-4 (and further in the ms): you studied one single trait, and it is not related to short-range endemism. So again, you cannot make any statement because 1 out of 3 species has juveniles preferring micro-habitats different from the adults.
L73: Are these 3 species phylogenetically or ecologically related? Why testing 3 species? What are your expectations if they have different life history traits?
L151-158: no figure, no replication, no test of significance…
L163: How can you conclude that? Invasive species can actually be positive for few species. You have no clue of any negative effect here (if the potential food estimate would have been replicated, there is very few evidence of food limitation in web-building / sit-and-wait spiders). The whole paragraph is completely hypothetical.
L185-225: out of scope. You should discuss the results instead.
L218-219: Again, you have no evidence of that.

Additional comments

Title: too vague, and does not reflect the content of the ms.
L6 (and further in the ms, e.g. L69): Do you make a difference between male and female mobility?
L13 (and further in the ms): You claim that the veld “can” make uniform landscapes, but does it? Where is actually the “landscape trap” here?
L31: Please explain how an established invasive plant can be seen as a disturbance.
L32-38: Paragraph poorly connected to the others, and not specific enough.
L52 (and further in the ms): do you consider mygalomorphs as species with limited dispersal? Do you test for pre-dispersal behavior in juveniles?
L65: what do you mean by “and other specialist species”?
L82-82: Reads like M&M
L89: I’m not sure I got it: SWA is not a biodiversity hotspot because it lost 90% of “original” habitat, right?
L102-103: Which mass?
Table 1 and in the text: nothing is said about statistics.
L123-144: could have been placed before.
L156 and below: better talk about taxonomic richness.
L173: Right, but you have no indication they did not…

---

## Round 0.2 · accepted · Accept

There are a few extra revisions you may consider incorporating below. Thank you for addressing previous suggestions so clearly.

# Reviewer 1 ·

Basic reporting

The authors have addressed my concerns.

Experimental design

The authors have addressed my concerns.

Validity of the findings

The authors have addressed my concerns.

Additional comments

The authors have addressed my concerns regarding the previous version of the manuscript. Please note the following minor suggestions:

line 94 (last paragraph of Intro): It is unnecessary to say "microhabitat habitat preference..."

Study Species paragraph: Provide citations, where available, for information about the species, such as the inferred relationship based on genetic data and burrow morphology/behaviour.

line 136 in Microhabitat preference: remove to after Microhabitat variables